# Energy Evolution and Damage Characteristics of Rock Materials under Different Cyclic Loading and Unloading Paths

Bing Sun [1], Haowei Yang [1], Junwei Fan [1], Xiling Liu [2] and Sheng Zeng [3,*]

1. School of Civil Engineering, University of South China, Hengyang 421001, China
2. School of Resources and Safety Engineering, Central South University, Changsha 410083, China
3. School of Resources Environment and Safety Engineering, University of South China, Hengyang 421001, China
* Correspondence: usczengs@126.com

**Abstract:** In order to study the deformation and failure characteristics of rocks under different cyclic loading and unloading paths, three stress path tests were conducted, and acoustic emission (AE) monitoring was conducted simultaneously. The mechanical characteristics and AE characteristics under different stress paths were analyzed, and the influences of the different stress paths on the energy dissipation and deformation damage were investigated. The law of energy evolution considering viscoelasticity under different stress paths was obtained. The concept of ultimate damage energy and its calculation method was proposed. The results show that the "hardening effect" of sandstone and granite under the constant lower limit (CLLCL) is the most significant in maximizing the mechanical property. The CLLCL imparts a stronger elastic property to rocks than the variable lower limit (VLLCL) does, while the VLLCL causes more damage to rocks than the CLLCL. A significant linear relationship between the proportion of damage energy and the proportion of elastic energy was discovered. Based on this linear relationship, the ultimate damage energy can be calculated for sandstone and granite. The evolution of the damage variable based on damage energy was compatible with the real damage condition, which validates the ultimate damage energy calculation method. The research results lay a theoretical foundation for the design and construction of geotechnical engineering.

**Keywords:** rock mechanics; cyclic loading and unloading; acoustic emission; energy evolution; damage characteristic; ultimate damage energy

## 1. Introduction

In geotechnical engineering activities, such as chamber excavation and tunnel boring, the surrounding rock containing a lot of joints and fissures is often in a state of cyclic loading and unloading, which may cause instability and a loss of employees and property [1–4]. When different excavation methods are applied, rocks are placed under different cyclic loading and unloading stress paths, leading to significant differences in the characteristics of rock failure [5]. The essence of rock failure, from a thermodynamic perspective, is the instability caused by energy evolution, and the deformation and failure of loaded rocks is an irreversible process of energy dissipation [6]. Consequently, studying the energy evolutions of rocks under different cyclic loading and unloading stress paths from the viewpoint of energy is beneficial for revealing the essential characteristics of rock failure and is of great importance to the stability analysis of geotechnical engineering.

At present, the existing research on rock energy may be divided into two areas: energy accumulation and dissipation characteristics, and energy failure criteria. In terms of energy accumulation and dissipation characteristics, Meng et al. [7] investigated the effect law of lithology and loading rate on the energy evolution process of loaded rocks and found that the energy density of three types of rocks showed a nonlinear evolution law when stress

was under different loading and unloading rates. Zhao et al. [8] analyzed the energy of rocks with various height–diameter ratios and discovered that the stored elastic energy of rocks under the same load was directly related to the height–diameter ratio and inversely proportional to the energy storage limit. Liu et al. [9] studied the influences of cycle number and upper limit stress on the evolution of the dissipated energy of rocks and found that the dissipated energy per unit volume also showed an overall increasing trend with an increasing cycle number and stress level. In terms of energy failure criteria, Xie et al. [10] established rock strength and overall failure criteria based on the principle of energy dissipation and release. Gong et al. [11] proposed the calculation method of ultimate elastic energy at peak strength based on the linear energy storage law and established a new rockburst proneness criterion. Li et al. [12] revealed the characteristics of the shale energy conversion process and energy evolution under triaxial cyclic loading and established a rock strength failure criterion based on energy catastrophe. Munoz et al. [13] proposed a new brittleness index based on the fracture strain energy derived from stress–strain measurements of rock. The energy evolution of rock can truly and objectively reflect the deformation and failure process of rock [14–17]. Consequently, some researchers used the energy principle to establish damage models that revealed the characteristics of rock damage. Many researchers have argued that dissipated energy is the useful work that causes rock failure, and the damage variable of rock was established based on the dissipated energy [18–22]. In fact, during the loading process, friction between particles and viscosity between liquids make the rock show nonlinear hysteresis; thus, non-dry rock is a viscoelastic material [23–25]. If the viscoelasticity is not taken into account for non-dry rocks, then the calculation result of the damage variable will be larger than the real value [26]. Therefore, in order to reflect the real mechanical properties of rock, the viscoelasticity in rock should be considered. In addition, other researchers used the acoustic emission (AE) technique to study the deformation and failure characteristics during cyclic loading and unloading from a microscopic viewpoint. AE signals can reflect micro-crack development and change caused by the evolution of internal defects in rocks [27,28]. At present, there is abundant research on AE characteristics in the process of rock deformation and failure, including the evolution of AE parameters under uniaxial [29,30], triaxial [31,32], and shear [33,34] loading modes, and the relationship between stress, strain, AE parameters, and AE location before peak strength [35–37]. Utilizing AE monitoring technology effectively reveals the development process of rock cracks. By combining this with energy analysis, it is possible to forecast the condition and type of rock failure more accurately and reveal the mechanism of instability.

Currently, the impacts of different stress paths under cyclic loading and unloading are seldom considered in studies on the energy evolution of rock. This is crucial for the energy analysis of rock. Different loading and unloading paths have different effects on the energy evolution during rock deformation, and each deformation corresponds to one energy condition [38,39]. In addition, the viscoelasticity of rock has rarely been considered in energy analyses, and damping energy and damage energy within dissipative energy have rarely been distinguished. The damage energy is the useful work that causes rock failure; hence, it is crucial to calculate the damage energy at the peak strength, which cannot be calculated at present. In this paper, uniaxial cyclic loading and unloading tests under different stress paths were carried out. In order to improve the test procedure, AE monitoring was carried out at the same time to real-time monitor the microscopic defects of the rock materials. The characteristics of the mechanics, AE, and energy under different stress paths were investigated. Considering the viscoelasticity of rock, the dissipative energy was subdivided into damage energy and damping energy, and the energy evolution law was further analyzed. The concept of ultimate damage energy and its calculation method was proposed. In order to validate the calculation method of ultimate damage energy and study the rock damage characteristics, the damage variable was established based on the damage energy driving rock failure. The research results provide a theoretical

grounding for further revealing the damage, deterioration, and instability mechanisms of rock under cyclic loading and unloading.

## 2. Materials and Methods

### 2.1. Specimen Preparation

Sandstone and granite were selected for this test. These rock specimens were collected from the surrounding rock at a mining site in Hunan, which is often used in geotechnical engineering. According to the standard of the International Society for Rock Mechanics (ISRM), a cylinder of Φ50 mm × 100 mm was made. The discrepancy between the diameter and height of each rock specimen was less than 0.1 mm, the non-parallelism of the two end faces was less than 0.03 mm, and the margin of error in the vertical direction was between 0.25 and −0.25. In order to reduce the specimens' discreteness as much as possible, a non-metallic ultrasonic detector was used to measure the wave velocity of the rock specimens. The specimens were named "rock type", "stress path", and "number of specimens". The rock specimens of the prepared test are shown in Figure 1.

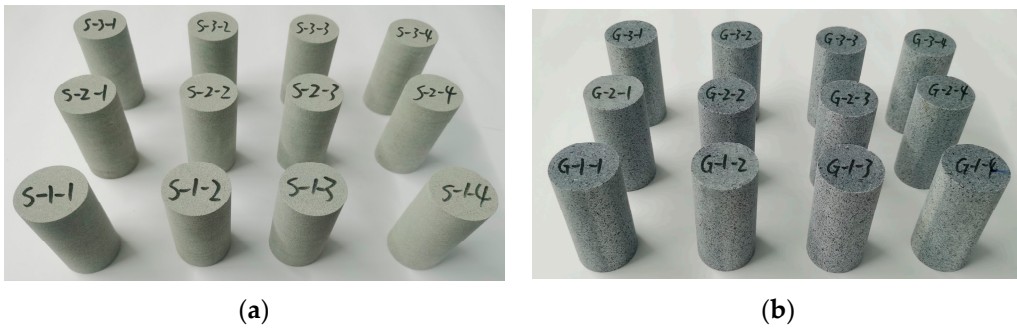

| (a) | (b) |

**Figure 1.** Rock specimens. (**a**) Sandstone; (**b**) granite.

### 2.2. Testing Schemes

A WAW-1000C electro-hydraulic servo press was used in the uniaxial compression and uniaxial loading and unloading tests. The piston displacement ranged from 0 to 800 mm, the maximum bearing capacity was 1000 kN, and the measurement error was ±1%. Stress and strain data were able to be recorded in real-time. A DS5-8B acoustic emission monitoring system was utilized to automatically monitor and collect the AE signals. The system is sensitive to AE signals and offers a wide frequency response range of 1 kHz–3 MHz. The sensor's acquisition rate was set to 1 MHz, and its threshold value was 40 dB. Eight AE sensors were equally distributed on the surface of each cylindrical specimen symmetrically with respect to the longitudinal axis. The schematic diagram of testing is shown in Figure 2.

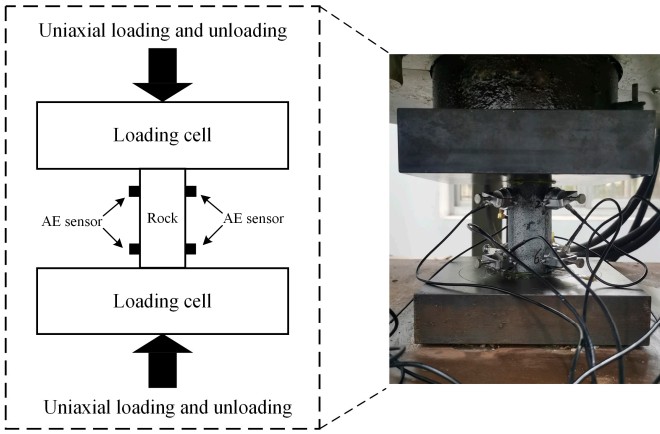

**Figure 2.** Schematic diagram of testing.

In order to simulate the stress and deformation characteristics of a foundation rock mass subjected to the disturbance of seismic waves or construction, three stress paths were designed: uniaxial compression (UC), constant lower limit cyclic loading and unloading (CLLCL), and variable lower limit cyclic loading and unloading (VLLCL):

(1) UC: Apply a 1 kN preload to the rock specimen and then load at a 1 kN/s loading rate until the rock specimen failure.

(2) CLLCL: Apply a 1 kN preload to the rock specimen and then load and unload at a $\pm 0.5$ MPa/s loading rate. When the load reaches 10%, 20%, 30%, 40%, 50%, 60%, 70%, 80%, and 90% of uniaxial compressive strength $\sigma_c$, it should be unloaded to 1 kN each time. After the ninth unloading, it should be directly loaded until the rock specimen failure. The stress path of CLLCL is shown in Figure 3.

(3) VLLCL: Apply a 1 kN preload to the rock specimen and then load and unload at a $\pm 0.5$ Mpa/s loading rate. When the load reaches 10%, 20%, 30%, 40%, 50%, 60%, 70%, 80%, and 90% of uniaxial compressive strength $\sigma_c$, unload it to 1 kN for the first time and then each subsequent unloading should be up to the maximum value of the previous loading. After the ninth unloading, it should be directly loaded until the rock specimen failure. The stress path of VLLCL is shown in Figure 4.

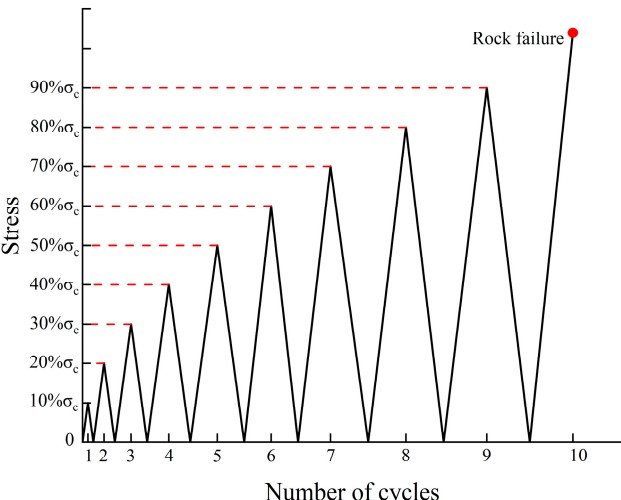

**Figure 3.** The stress path of constant lower limit cyclic loading and unloading (CLLCL).

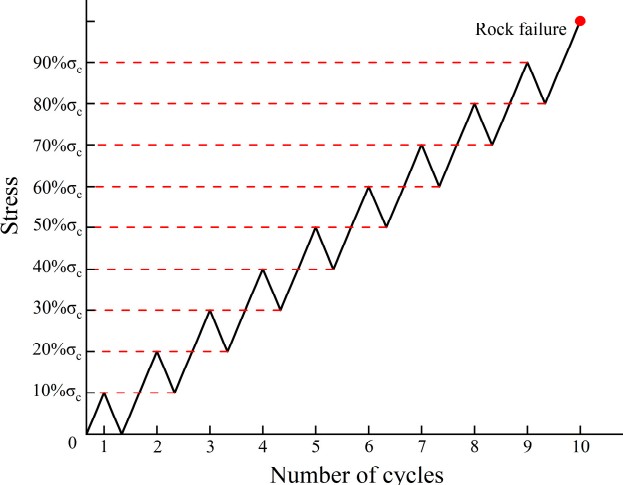

**Figure 4.** The stress path of variable lower limit cyclic loading and unloading (VLLCL).

## 3. Testing Results

### 3.1. Stress–Strain Characteristics

The stress–strain behaviors of rock describe the intrinsic physical and mechanical characteristics. The stress–strain curves for sandstone and granite under CLLCL are shown in Figure 5. The curve of stress–strain varies with the stress level. When the stress level at the point of unloading is low, the unloading curve basically returns to the origin, demonstrating elastic recovery. On the contrary, when the stress level at the unloading point is high, the unloading curve deviates from the original loading curve, resulting in irreversible plastic deformation. The unloading curve no longer returns to the origin but instead forms a hysteresis loop, which smoothly intersects the loading curve. Under UC and CLLCL, the stress–strain curves of the rock specimens exhibit a similar changing trend. They are quite close throughout the phases of compaction and elasticity, and the differences are tiny during the plastic stage. The stress sharply drops after it reaches the peak point, and the rock loses all its bearing capacity. The envelopes of the stress–strain curves under CLLCL and UC are basically coincided, which reflects the "memory effect" of rock materials. The average peak strengths of the rock specimens under two types of cyclic loading and unloading improve by 4.61% compared with $\sigma_c$, which shows a strength increase.

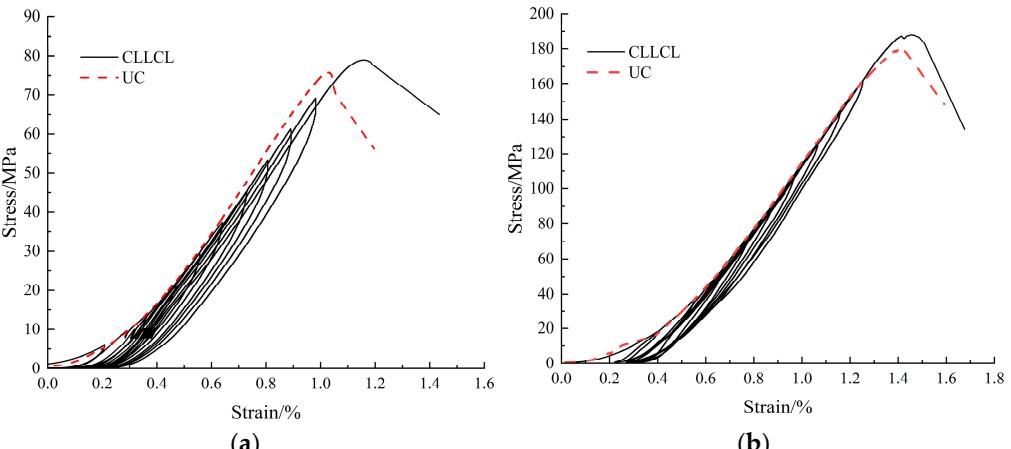

**Figure 5.** Stress–strain curves under CLLCL. (**a**) Sandstone; (**b**) granite.

The stress–strain curves for sandstone and granite under VLLCL are shown in Figure 6. The stress–strain relationship varies with the stress level. As the loading and unloading amplitude at each level of the cyclic loading during VLLCL is only 10% of the $\sigma_c$, the strain created by loading and unloading is small. In the next loading process, the curve smoothly intersects with the previous unloading curve, forming a tiny hysteresis loop. Under UC and VLLCL, the stress–strain curves for the rock specimens exhibit a similar changing trend. Compared with CLLCL, the envelope of the curve under VLLCL is more consistent with the curve under UC, indicating that the "memory effect" of rock under VLLCL is more significant. The peak strengths of sandstone and granite under VLLCL are 3.90% greater than the $\sigma_c$, indicating that the rock strength under the stress path also increases but is not as significant as when under CLLCL.

### 3.2. Elastic Modulus Evolution

The elastic modulus (Young's modulus) is an important parameter for rock deformation characterization, reflecting the deformation resistance of loaded rock specimens. In order to compare the deformation resistance of rock under uniaxial compression and cyclic loading and unloading, stress interval data points within the range of 30~70% at each loading and unloading level of the curve were linearly fitted, and, within the corresponding range, the stress interval data points of uniaxial compression were also linearly fitted. This is shown in Figure 7.

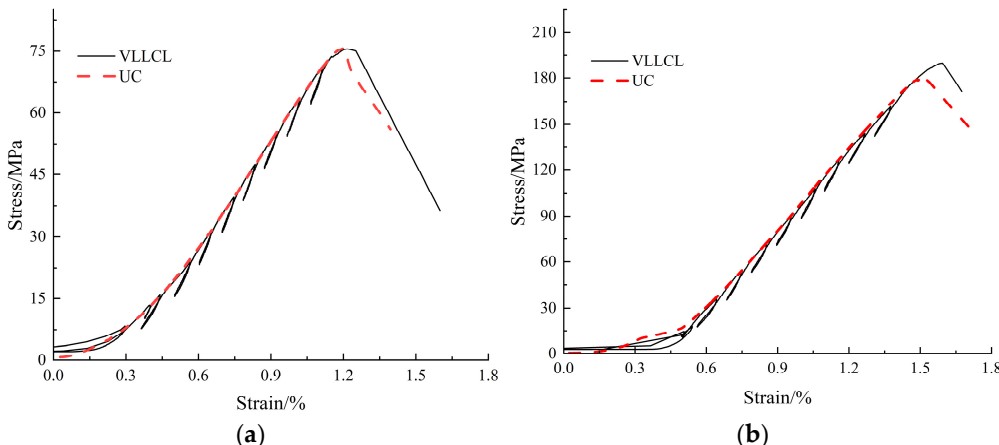

**Figure 6.** Stress–strain curves under VLLCL. (**a**) Sandstone; (**b**) granite.

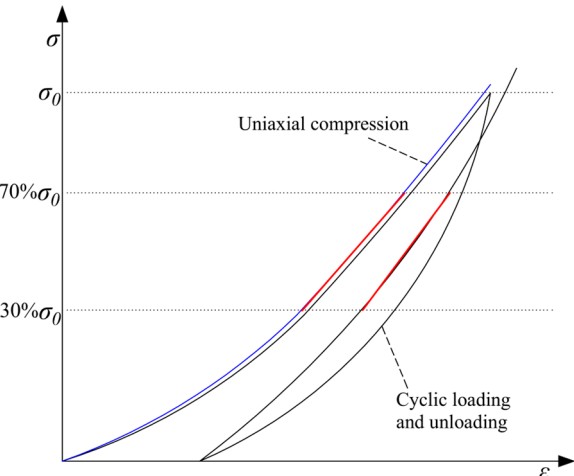

**Figure 7.** The calculation method of modulus.

The relationship between the moduli of sandstone and granite and the number of cycles is shown in Figure 8. Under different stress paths, the modulus of sandstone ranges from 2.3 GPa to 13.4 Gpa, and the modulus of granite ranges from 3.1 Gpa to 23.6 Gpa, showing that granite is denser than sandstone and has more deformation resistance. The modulus evolutions of sandstone and granite under different stress paths have similar laws. In the late loading and unloading period, the elastic modulus of the two rocks under CLLCL still tends to increase. In the same period, under UC and VLLCL, the modulus curves of granite have a downward trend, while the curves of sandstone become flat and are close to the lowest value. Such a difference between granite and sandstone is due to the fact that the compactness of granite is better than sandstone. The moduli under cyclic loading and unloading are greater than under UC, indicating that the cyclic loading and unloading stress paths strengthen the rock deformation resistance. As the number of cycles increases from one to seven, the elastic moduli increase continually, yet the elastic moduli under VLLCL are always more than the elastic moduli under CLLCL and UC. After the seventh cycle, the elastic moduli under UC and VLLCL decrease exponentially, suggesting that the rocks reach the plastic stage and abundant micro-cracks start to develop, expand, and connect. However, the elastic moduli under CLLCL do not decrease at this stage; rather, they tend to gradually increase and exceed under VLLCL in the tenth cycle when rock failure is about to be observed. This demonstrates that the stress path not only increases the elastic moduli but also prolongs the elastic deformation stage, thus improving the mechanical properties.

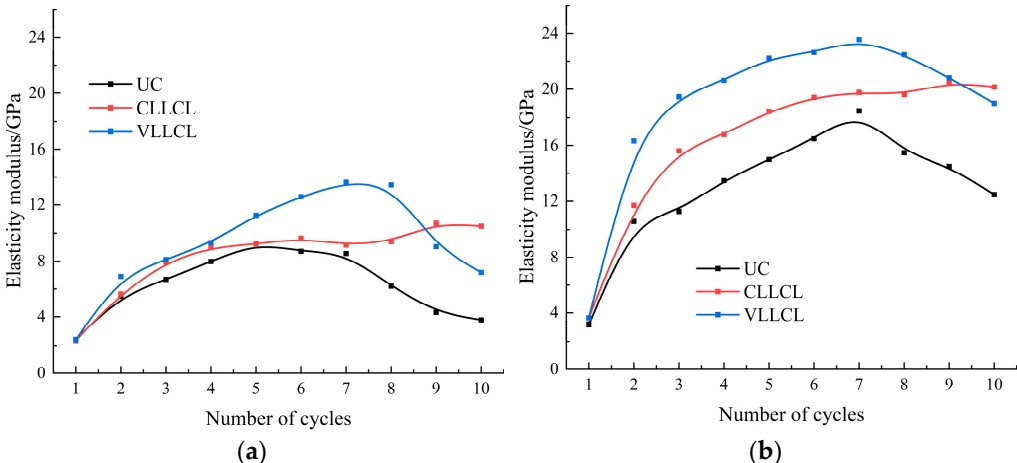

**Figure 8.** The relationship between the elastic modulus and the number of cycles. (**a**) Sandstone; (**b**) granite.

The above research reveals that, compared with UC, the strength and modulus of rock under cyclic loading and unloading are increased. It can be inferred that the cyclic loading and unloading stress path improves the mechanical properties of sandstone and granite. This is known as the "hardening effect" of rock under cyclic loading and unloading. Thanks to the "hardening effect", newly generated cracks are filled with rock debris, increasing the friction force among fissure planes and improving the strength and elastic properties of the rock [40].

### 3.3. AE Characteristics

AE during rock loading is irreversible. AE sources not only indirectly reflect the deformation and failure process of rock but also point out the spatial locations of micro-crack initiation, propagation, and penetration [41,42]. The AE energy evolutions and locations of sandstone and granite under UC are shown in Figure 9. The characteristics of the AE energy evolution are used to classify the AE energy evolution process into three periods: the small increase period (before point A), the relatively quiet period (AB), and the active period (after point B). During the small increase period, the micro-cracks, joints, and pores of the rock are closed, the AE energy increases slightly, and AE sources are distributed randomly in the specimen. In the early relatively quiet period, the pressure on the rock specimens is insufficient to cause the formation of new cracks, resulting in little AE energy activeness. Essentially, the cumulative AE energy curves are horizontal. The number of AE location points increase slightly at about 40% $\sigma_P$ (peak stress), indicating crack initiation. In the late relatively quiet period, AE location points gradually gather together and connect at about 80% $\sigma_P$, cracks begin to propagate, and small damages accumulate. In the active period, the number of AE location points increase dramatically, indicating continuous crack expansion and penetration accompanied by the generation of a large number of new cracks, and the AE source is extremely active. The cumulative AE energy curves show a sudden increase change approximate to a vertical line.

The AE energy evolutions and locations of sandstone and granite under CLLCL and VLLCL are shown in Figures 10 and 11. The characteristics of the AE energy evolution also are used to classify the AE energy evolution process into three periods: the small increase period (before point A), the relatively quiet period (AB), and the active period (after point B). During the small increase period, AE sources are distributed randomly in the specimens, the micro-cracks, joints, and pores of the rock are closed, and the AE energy increases slightly. In contrast to that during UC, the AE energy under cyclic loading and unloading exhibits a progressive increase in the relatively quiet period as the number of cycles increases. In this period, because the grains of sandstone are loose, the sandstone debris generated in the last loading are loose and rub one another as soon as the next

loading begins, so a small amount of high-energy AE energy under CLLCL is generated before loading turns to unloading and vice versa [43]. The sandstone under VLLCL does not generate high-energy AE energy. This is because the unloading amplitude of VLLCL is less than that of CLLCL, and the debris generated by the sandstone is not enough to be loose and rub one another, so no high-energy AE energy is generated. In the fourth cycle, the number of AE location points increases slightly, indicating crack initiation. During the ninth cycle, the AE location points start to gather and begin to be closely connected as new micro-cracks are created and the already existing ones are extended.

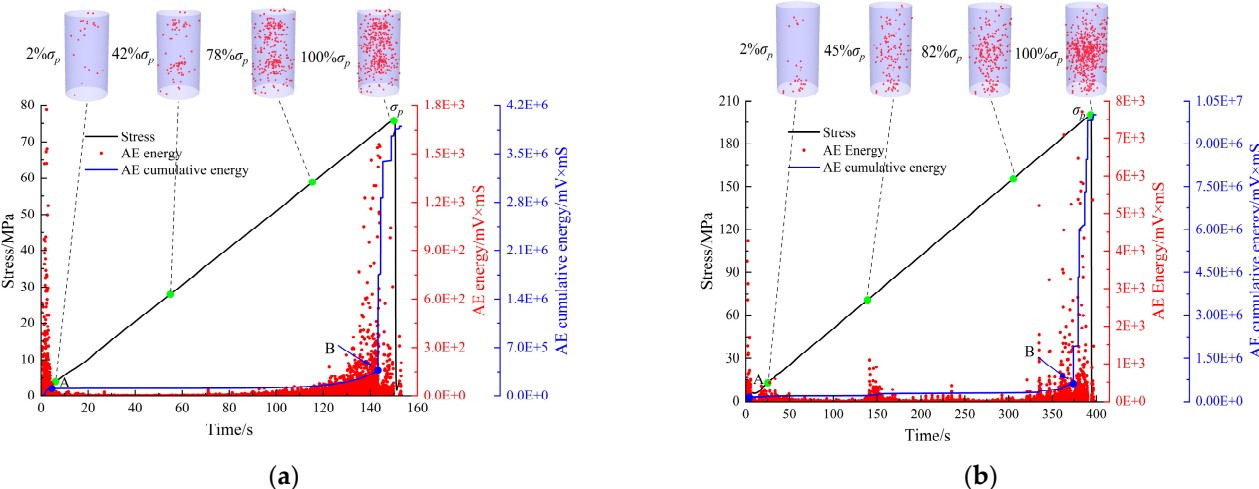

**Figure 9.** The AE energy evolution and location under UC. (**a**) Sandstone; (**b**) granite.

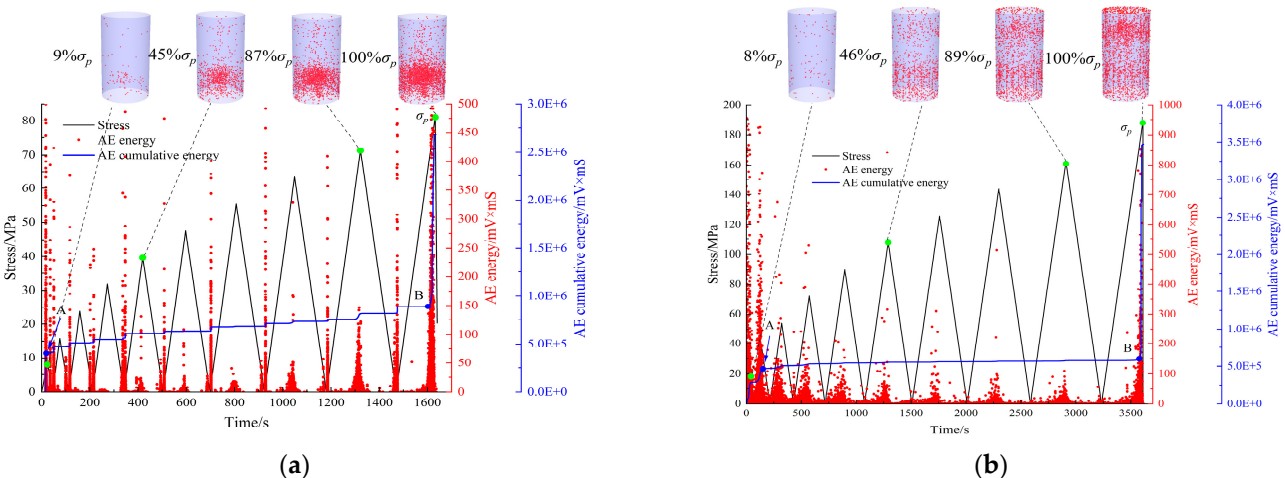

**Figure 10.** The AE energy and location evolution under CLLCL. (**a**) Sandstone; (**b**) granite.

In the active period, the number of AE location points increase dramatically, the cracks expand and penetrate, accompanied by the generation of a large number of new cracks, and the AE source is extremely active. The cumulative AE energy curves show a sudden increase that approximates a vertical line.

The proportions of AE energy in the three AE periods under different stress paths are shown in Table 1. Under different stress paths, most of the AE energy is distributed in the active periods, the proportions of which are 91.3%, 60.4%, and 81.8%, respectively, and only a small part is distributed in the first two periods. This demonstrates that the failures of sandstone and granite show the burstiness. However, compared with UC, the cyclic loading unloading stress path inhibits the burstiness, and CLLCL has the strongest inhibition effect.

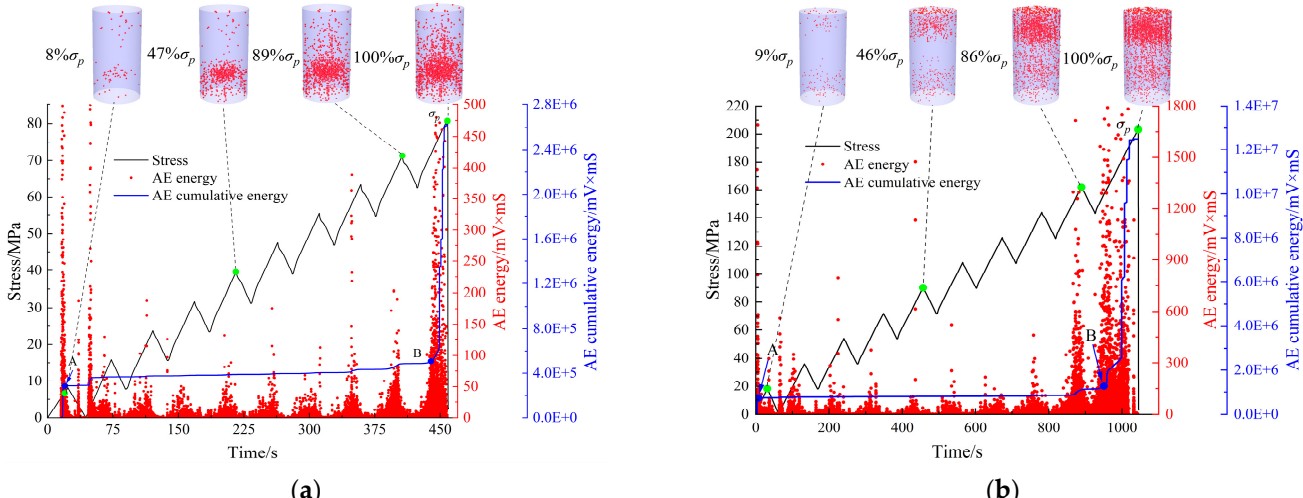

**Figure 11.** The AE energy evolution and location under VLLCL. (**a**) Sandstone; (**b**) granite.

**Table 1.** The proportions of AE energy in AE period under different stress paths.

| Stress Path | Small Increase Period | Relative Quiet Period | Active Period |
|:-----------:|:---------------------:|:---------------------:|:-------------:|
| UC | 1.8% | 6.9% | 91.3% |
| CLLCL | 14.2% | 25.4% | 60.4% |
| VLLCL | 6.6% | 11.6% | 81.8% |

## 4. Energy Evolution

### 4.1. Traditional Energy Transformation Theory

Rock failure is an instability phenomenon driven by energy. Low energy (sound energy, heat energy, radiant energy, etc.) in energy conversion can be ignored. According to the first law of thermodynamics, input energy $U_i$ is transformed into elastic energy $U_i^e$ and dissipated energy $U_i^d$ during cyclic loading and unloading. The formula for calculating energy $U_i$, $U_i^e$, and $U_i^d$ is as follows [44]:

$$U_i = \int_{\varepsilon_0}^{\varepsilon_A} \sigma_i^+ d\varepsilon \tag{1}$$

$$U_i^e = \int_{\varepsilon_C}^{\varepsilon_A} \sigma_i^- d\varepsilon \tag{2}$$

$$U_i^d = U_i - U_i^e = \int_{\varepsilon_0}^{\varepsilon_A} \sigma_i^+ d\varepsilon - \int_{\varepsilon_C}^{\varepsilon_A} \sigma_i^- d\varepsilon \tag{3}$$

where $\sigma_i^+$ and $\sigma_i^-$ are the loading stress function and unloading stress function in the *i*th cycle, respectively.

### 4.2. Energy Transformation Theory of Rock Viscoelasticity Is Considered

Rock is a material that is heterogeneous, discontinuous, and anisotropic. The friction between rock particles and the liquid's viscosity produces a nonlinear hysteresis effect in rock throughout the pressing process [26]. Consequently, it is required to consider the viscoelasticity of rock and to subdivide the dissipated energy into the damping energy needed to overcome the viscosity of rock and the damage energy used to initiate and propagate micro-cracks and plastic deformation in rock.

The energy calculation diagram is shown in Figure 12. At point *B*, the last unloading curve intersects the next loading curve, generating a closed hysteresis loop *BCB*. The viscosity and elasticity of the rock influence the geometry of the hysteresis loop. The resulting deformation is a viscoelastic deformation without plastic deformation. The elastic

energy is not lost, and the energy loss in this period is caused by the damping force [26,45]. Therefore, the area of the hysteresis loop *BCB* indicates the damping energy $U_i^{dz}$ dissipated to overcome the rock viscosity (including liquid viscosity and interface friction). The damage energy $U_i^{ds}$ causes rock damage, and it can be calculated by subtracting the damping energy from the dissipation energy, and its formula is [26,46]

$$U_i^{dz} = \int_{\varepsilon_C}^{\varepsilon_B} \left( \sigma_{i+1}^+ - \sigma_i^- \right) d\varepsilon \tag{4}$$

$$U_i^{ds} = U_i^d - U_i^{dz} = \int_{\varepsilon_0}^{\varepsilon_A} \sigma_i^+ d\varepsilon - \int_{\varepsilon_C}^{\varepsilon_A} \sigma_i^- d\varepsilon - \int_{\varepsilon_C}^{\varepsilon_B} \left( \sigma_{i+1}^+ - \sigma_i^- \right) d\varepsilon \tag{5}$$

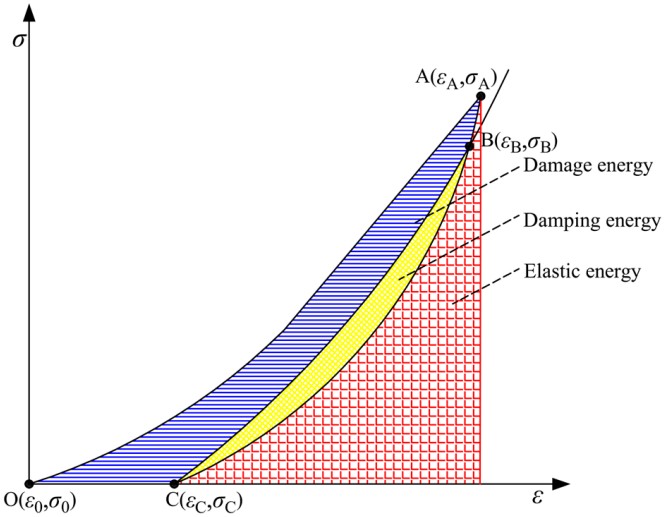

**Figure 12.** Diagram of energy calculation.

### 4.3. Analysis of Energy Evolution under Cyclic Loading and Unloading

The relationship between the energy (input energy, elastic energy, damage energy, and damping energy) of sandstone and granite and the number of cycles is shown in Figures 13 and 14. Figure 13 shows that, under CLLCL, the input energy and elastic energy of sandstone and granite increase continuously as the number of cycles increases, exhibiting evident nonlinear growth characteristics. The damping energy and damage energy increase relatively slowly, suggesting that there is less energy dissipation. Under CLLCL, as the number of cycles increases, the testing machine works on the rocks, leading to an increase in the input energy. A large proportion of the input energy is transformed into elastic energy and stored in the rock. A tiny proportion of the input energy is transformed into damping energy and damage energy, which are used to overcome the viscosity of rock and promote the start and development of micro-cracks, respectively.

It can be seen from Figure 14 that, under VLLCL, with an increase in the cycle number, the input energy and damage energy of the sandstone and granite first increase, then decrease, and then increase. The energy value in the second cycle is abnormal. This is due to the closure and compression of micro-cracks during the compaction stage, which increases rock deformation and requires more energy. As a result, compared with the first cycle, considerable increases in the input energy and damage energy are observed in the second cycle. The elastic energy increases as the number of cycles increases, showing that the elastic property of rock is improved; the damping energy decreases initially and tends to be stable after the second cycle. Under VLLCL, as the number of cycles increases, a large proportion of the input energy is transformed into damage energy, and a tiny proportion is transformed into elastic energy and damping energy. Thus, it can be concluded that in the three stress paths, the stress path of CLLCL makes the rock elastic property strongest, whereas the stress path of VLLCL causes the most damage to the rock.

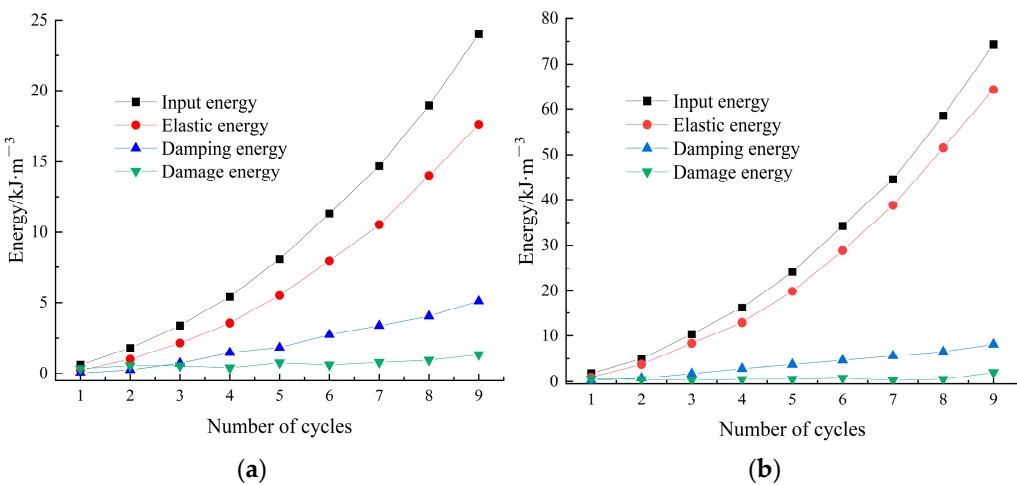

**Figure 13.** Rock energy evolution under CLLCL. (**a**) Sandstone; (**b**) granite.

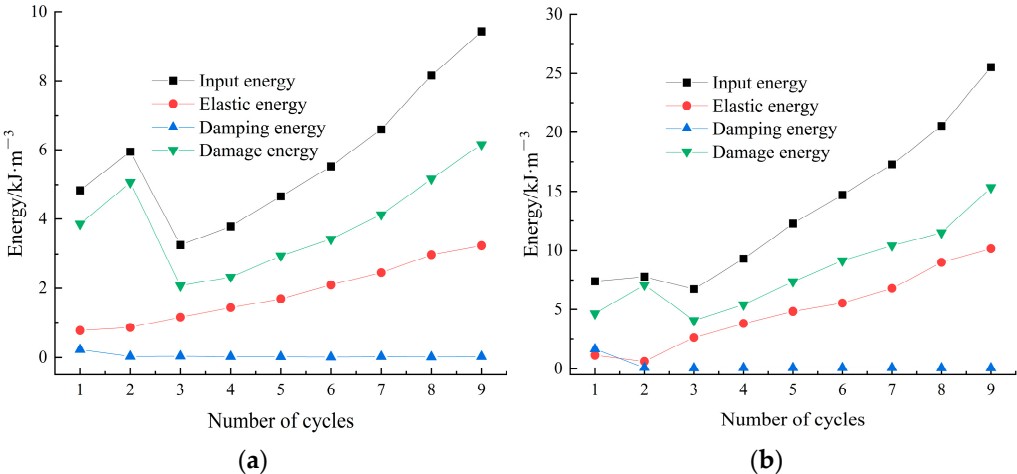

**Figure 14.** Rock energy evolution under VLLCL. (**a**) Sandstone; (**b**) granite.

With an increase in the number of cycles, the loading force increases correspondingly, which changes the internal structure of the rock and the energy distribution. To further examine the energy distribution and evolution law of rocks at each cycle level, the proportions of elastic energy, damage energy, and damping energy in the input energy at each cycle level were calculated, as seen in Figures 15 and 16. It can be seen from Figure 15 that the proportion of elastic energy in sandstone and granite under CLLCL increases and finally increases to 73.27% and 85.32%, respectively. The proportion of damage energy increases as the number of cycles increases, whereas the proportion of elastic energy decreases, showing a reverse trend. This is because the rock deformation is obvious during the compaction stage, resulting in a greater damage energy, but the energy dissipation in the latter stage is mainly damping energy, while damage energy is at a very low level. The proportion of damping energy in sandstone increases first and then stabilizes, but in granite, it increases initially and then decreases gradually after reaching its maximum value in the fourth cycle. This is because granite is denser than sandstone, resulting in less debris during loading and unloading. In the later stages of loading and unloading, the debris and fractures are gradually compressed, and the amount of damping energy needed to overcome the fracture interface's friction is gradually reduced.

Figure 16 demonstrates that under VLLCL, the proportion of damage energy in sandstone and granite increases and then decreases and finally stabilizes at 61.35~65.34% and 56.02~61.9%, respectively. The proportion of damping energy decreases first and then stabilizes at 0.13~0.45%. The proportion of elastic energy and the proportion of damage energy show a reverse trend, with the proportion of elastic energy increasing initially and

then stabilizing at 34.39~41.29%. The evolution curve of rock energy proportion can reflect the energy distribution at each level of each cycle. Under CLLCL, elastic energy dominates each cycle, whereas damage energy dominates each cycle under VLLCL.

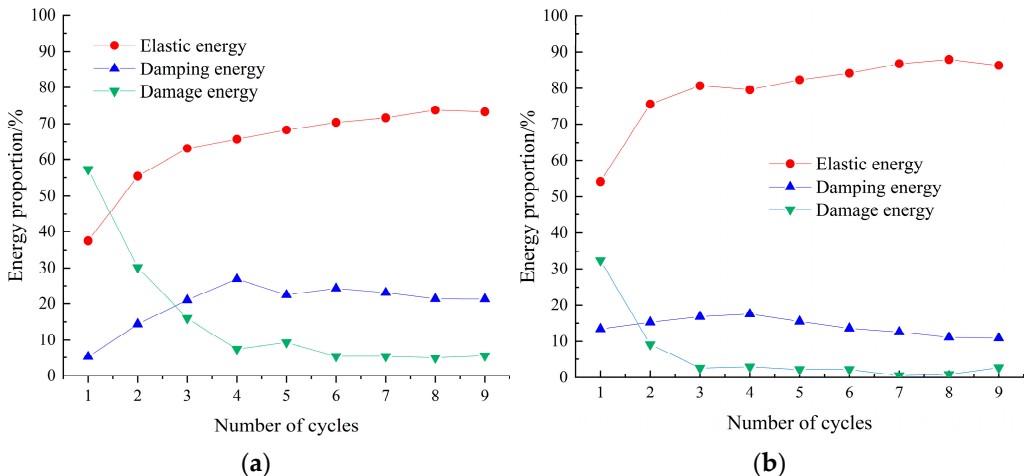

**Figure 15.** The evolution of rock energy proportion under CLLCL. (**a**) Sandstone; (**b**) granite.

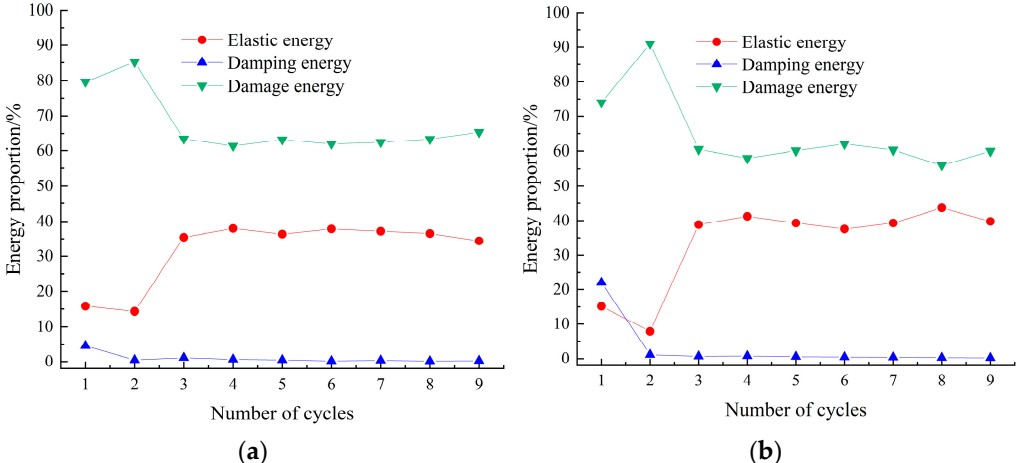

**Figure 16.** The evolution of rock energy proportion under VLLCL. (**a**) Sandstone; (**b**) granite.

## 5. Ultimate Damage Energy

### 5.1. The Concept and Calculation Method of Ultimate Damage Energy

The damage energy in rock energy is the useful work that drives the failure of rock; hence, it is crucial to calculate the damage energy at the peak strength in order to predict rock failure. Nevertheless, there is currently no method for calculating the peak damage energy. Due to the heterogeneity and brittleness of rock, the strength of each rock specimen cannot be determined in advance. As a result, the unloading test of rock specimens at the peak strength is impossible, and the damage energy at the peak strength cannot be calculated. In order to solve this problem, the concept of ultimate damage energy is proposed for the first time, which is defined as the damage energy at the peak strength of rock. Based on Section 4.1's study of energy evolution, it is known that the proportions of damage energy and elastic energy of rock under CLLCL or VLLCL show a reverse trend. Therefore, a linear fitting of the two kinds of energy proportion data was carried out to try to determine the relationship between the energy of each part in sandstone and granite, and then the ultimate damage energy was derived. The fitting results are shown in Figures 17 and 18.

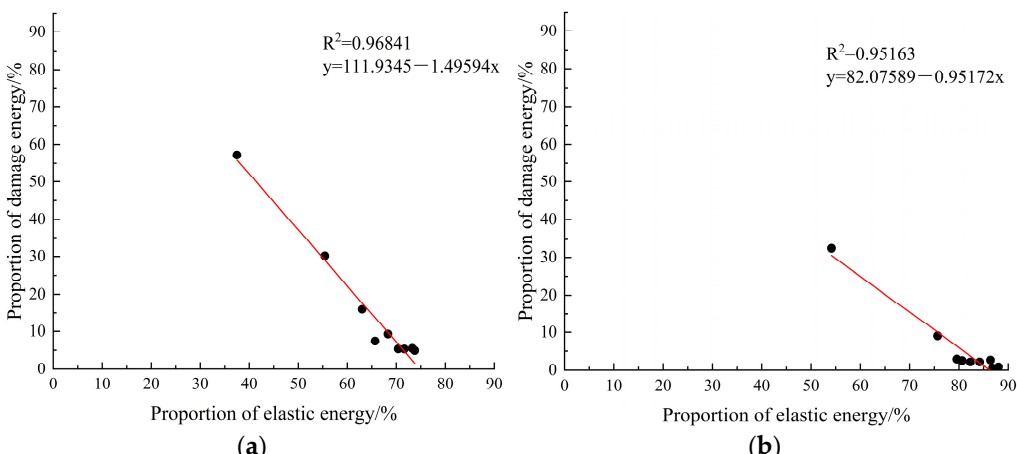

**Figure 17.** Fitting between the proportion of damage energy and proportion of elastic energy under CLLCL. (**a**) Sandstone; (**b**) granite.

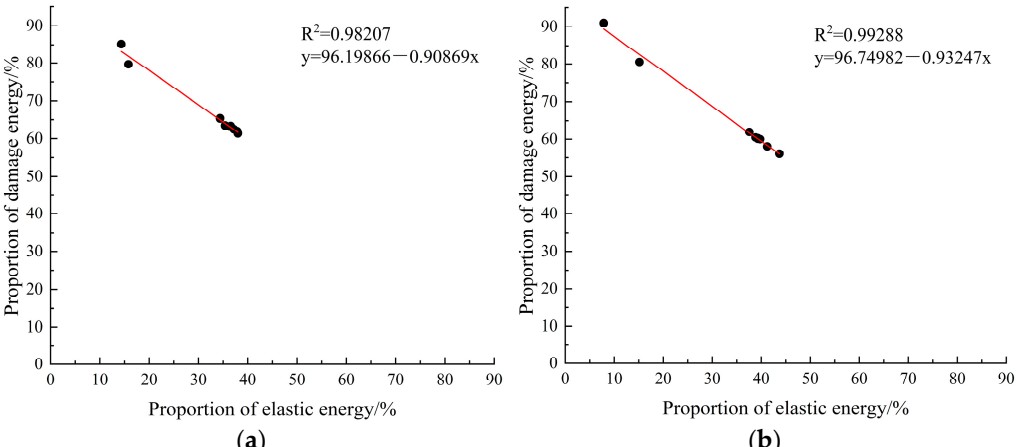

**Figure 18.** Fitting between the proportion of damage energy and proportion of elastic energy under VLLCL. (**a**) Sandstone; (**b**) granite.

Figures 17 and 18 reveal that the $R^2$ of the fitting curves is larger than 0.95, suggesting an adequate data fitting. Consequently, the linear fitting formula for the elastic energy proportion and the damage energy proportion can be calculated as follows:

$$\frac{U^{ds}}{U} = b + a\frac{U^e}{U} \tag{6}$$

where $a$ and $b$ are fitting parameters.

The relationships between damage energy, elastic energy, and input energy can be obtained from the transformation of Equation (6):

$$U^{ds} = bU + aU^e \tag{7}$$

Substituting fitting coefficients $a$ and $b$ into Equation (7) yields the following relationships between damage energy, elastic property, and input energy for sandstone and granite under CLLCL and under VLLCL:

$$\begin{cases} U^{ds} = 111.9345\%U - 1.49594U^e \\ U^{ds} = 82.07589\%U - 0.95172U^e \\ U^{ds} = 96.19866\%U - 0.90869U^e \\ U^{ds} = 96.74982\%U - 0.93247U^e \end{cases} \tag{8}$$

Elastic energy at peak strength can be expressed as [20]

$$U^e = \frac{\sigma_0}{2E_0} \tag{9}$$

where $\sigma_0$ is the peak stress, and $E_0$ is the initial elastic modulus.

The simultaneous Equations (8) and (9) can be used to calculate the damage energy of rock at the peak strength (the ultimate damage energy). The ultimate damage energies of the rock specimens under different stress paths are shown in Table 2.

**Table 2.** The ultimate damage energies under different stress paths.

| Stress Path | Rock Type | Ultimate Damage Energy/kJ·m$^{-3}$ |
|:---:|:---:|:---:|
| CLLCL | Sandstone | 3.84 |
| | Granite | 4.16 |
| VLLCL | Sandstone | 7.01 |
| | Granite | 16.64 |

*5.2. Damage Evolution Based on Damage Energy*

Some researchers use the normalized dissipated energy to describe rock damage [18–20]. The former parts of this paper demonstrate, however, that the internal cause of plastic deformation in rock and promotion of micro-crack initiation and propagation is damage energy. Therefore, it is better to describe the damage evolution of rock based on the normalized damage energy.

Damage variable $D$ [26] based on normalized damage energy can be expressed as

$$D_i = \frac{U_i^{ds}}{U^{ds}} \tag{10}$$

where $D_i$ is the $i$th cycle's damage variable, $U_i^{ds}$ is the $i$th cycle's cumulative damage energy, and $U^{ds}$ is the total damage energy. When $D = 0$, the rock is undamaged, and when $D = 1$, the peak strength has been reached and rock failure occurs.

The damage evolutions of sandstone and granite under CLLCL and VLLCL are shown in Figure 19. The damage evolution of rock under different stress paths can be divided into two stages: stable accumulation and accelerated accumulation. In the stage of stable accumulation, the damage variable increases linearly with the number of cycles, and damage accumulation is slow. Under VLLCL, the initial damage and damage development rates of rock are somewhat greater than when rock is under CLLCL, resulting in a bigger damage variable. In the seventh cycle, it moves to the accelerated damage accumulation stage, wherein the rate of damage increases significantly and reaches the maximum value. Rock failure can almost be observed at this stage. Due to the accumulation of early damage, the damage of rock under VLLCL is greater than that under CLLCL at this stage. However, the increased rate of the damage variable under CLLCL is larger, indicating that the rate of micro-crack propagation and penetration is larger at this moment, and the macro-failure surface forms extremely rapidly. Throughout the entire damage evolution stage of rock, the damage variable under VLLCL is always greater than that under CLLCL, indicating that the stress path of VLLCL causes more damage to rock and a more pronounced decline in rock mechanical properties. This is consistent with the energy analysis conclusion that the stress path of VLLVL causes more damage to the rock. The damage characteristics were compatible with the real damage conditions in the rocks, which validates, to a certain degree, the validity of the calculation method for ultimate damage energy.

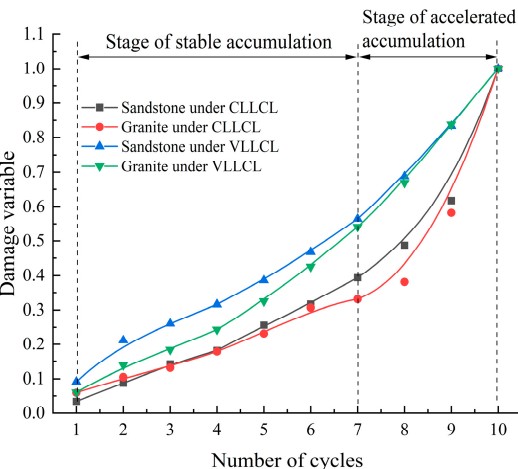

**Figure 19.** The damage evolution under different stress paths.

## 6. Conclusions

(1)  Due to the "hardening effect", the strength of cyclic loading and unloading was slightly greater than that of uniaxial compression. The "hardening effect" of sandstone and granite under CLLCL was the most significant, maximizing the mechanical property.

(2)  Under different stress paths, the failures of sandstone and granite showed burstiness. Compared with UC, the cyclic loading–unloading stress paths had a certain degree of inhibition in the burstiness, and CLLCL had the strongest inhibition. The combination of AE source location and AE parameters can well reflect the micro-crack initiation, propagation, and penetration in the process of rock loading.

(3)  Each energy exhibited nonlinear evolution characteristics as the number of cycles increased. Under CLLCL, the elastic energy dominated, which illustrates sandstone and granite under CLLCL as having stronger elastic properties than when under VLLCL; under VLLCL, the damage energy dominated, which illustrates VLLVL as causing more damage to rock than CLLCL.

(4)  A strong linear relationship between the proportion of damage energy and the proportion of elastic energy was found. Based on this linear relationship, the concept of ultimate damage energy and its calculation method were proposed, which can solve the problem of the inability to calculate the damage energy at the peak strength.

(5)  The damage evolution curve based on damage energy can be divided into two stages: stable accumulation and accelerated accumulation. Throughout the whole damage evolution process, VLLCL caused more damage to the rocks than CLLCL. The damage characteristics were compatible with the damage conditions in the rocks, which validates, to a certain degree, the validity of the calculation method for ultimate damage energy.

**Author Contributions:** Conceptualization, B.S. and S.Z.; methodology, B.S. and H.Y.; validation, S.Z.; formal analysis, H.Y.; investigation, X.L. and J.F.; resources, B.S. and S.Z.; writing—original draft preparation, H.Y. and B.S.; writing—review and editing, H.Y. and S.Z.; visualization, H.Y.; supervision X.L. and J.F.; funding acquisition, B.S. All authors have read and agreed to the published version of the manuscript.

**Funding:** This work was supported by the Natural Science Foundation of Hunan Province, China (grant no. 2021JJ30575) and the National Natural Science Foundation of China (grant no. 51204098).

**Institutional Review Board Statement:** Not applicable.

**Informed Consent Statement:** Not applicable.

**Data Availability Statement:** Some or all data that support the findings of this study are available from the corresponding author upon reasonable request.

**Conflicts of Interest:** The authors declare no conflict of interest.

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
