# Peer review of "Energy Evolution and Damage Characteristics of Rock Materials under Different Cyclic Loading and Unloading Paths"

_buildings, doi:10.3390/buildings13010238_

Round 1
Reviewer 1 Report
In this paper, the energy evolution of rock under different stress paths was studied, and the concept and calculation method of ultimate damage energy were proposed. The work has certain innovations and significance. However, there are the following problems that should be paid attention to and revised in the manuscript.
(1) In Figure 1, the meaning of the number of each rock specimen needs to be explained. For example, what does “S-1-1” mean?
(2) In Figure 7, the author marked the position of “30%σ” and “70%σ”, but did not mark the specific position of “σ”.
(3) In Section 3.2, the expression of modulus is ambiguous. The concepts of modulus in uniaxial compression and modulus in cyclic loading and unloading should be distinguished so that readers can understand them clearly.
(4) Please explain why the end of the stress-strain curve under UC in Figure 8 (a) becomes flat, while the end of the stress-strain curve under UC in Figure 8 (b) continues to decline.
(5) The author needs to check the collocation of words in the full text. For example, it is more appropriate to replace “elastic characteristic” with “elastic property” in Section 4.1.
(6) The latest references of AE characteristics of rock need to be added in the introduction.
Author Response
Point 1: In Figure 1, the meaning of the number of each rock specimen needs to be explained. For example, what does “S-1-1” mean?
Response 1: We added the numbering rules of rock samples in line 101 under the "track changes".
Point 2: In Figure 7, the author marked the position of “30%σ” and “70%σ”, but did not mark the specific position of “σ”.
Response 2: We replaced the previous “σ” with “σ0” and marked the position of “σ0” in Figure 7.
Point 3: In Section 3.2, the expression of modulus is ambiguous. The concepts of modulus in uniaxial compression and modulus in cyclic loading and unloading should be distinguished so that readers can understand them clearly.
Response 3: We appreciate your good advice. In Section 3.2, we have made a comprehensive modification to distinguish the deformation modulus of uniaxial compression and the elastic modulus of cyclic loading and unloading.
Point 4: Please explain why the end of the stress-strain curve under UC in Figure 8 (a) becomes flat, while the end of the stress-strain curve under UC in Figure 8 (b) continues to decline.
Response 4: This is due to the properties of the two kinds of rocks. The compactness of granite is better than that of sandstone, and the modulus still has a downward trend in the late loading and unloading period, while the elastic modulus curve of sandstone has become flat since it is close to the lowest value. In lines 163-165 under the "track changes", we have been modified and added to the reviewer's suggestions.
Point 5: The author needs to check the collocation of words in the full text. For example, it is more appropriate to replace “elastic characteristic” with “elastic property” in Section 4.1.
Response 5: Thank you for your good suggestions. We have made corrections according to the reviewer’s comments in line 295 under the "track changes", and the full text has been polished.
Point 6: The latest references of AE characteristics of rock need to be added in the introduction.
Response 6: Thank you for your good suggestions. We reviewed the AE characteristics of rock in the Introduction, and relevant references have been adjusted and added in lines 71-73 under the "track changes".
See the attachment for details

Reviewer 2 Report
The subject of the manuscript belongs to geotechnical engineering.
Extensive editing improvement is needed.
State clearly what is new in your research. How the addition of acoustic emission monitoring has imroved the testing procedure? Your conclusion on rock hardening (instead of strenghtening) of cyclic loading is based on testing sandstone and granite. Is it reasonable a generalization of that effect?
Have you any indication of locations of fracture initiation based on acoustic emission sources in the specimens?
Are your estimation and calulculations of energy referred to the whole volume of the specimen? Is it possible for an estimation of energy evolution at the fracture initiation locations?
Improve the conclusions discussing what is your contribution.
Author Response
Point 1: The subject of the manuscript belongs to geotechnical engineering.
Response 1: We gratefully appreciate for your valuable suggestion. We are very sorry for our incorrect writing. In lines 31-42 under the "track changes", we have revised the background in the Introduction. In lines 26-27 under the "track changes", we emphasized the contribution of this manuscript to geotechnical engineering.
Point 2: Extensive editing improvement is needed.
Response 2: We thank the reviewer for pointing out this issue. We have improved the editing of the full text, including polishing the language and combing the sentence structure to improve the readability and logic of the manuscript.
Point 3: State clearly what is new in your research. How the addition of acoustic emission monitoring has imroved the testing procedure? Your conclusion on rock hardening (instead of strenghtening) of cyclic loading is based on testing sandstone and granite. Is it reasonable a generalization of that effect?
Response 3: Thank you for the above suggestions. In this paper, our main innovation is “the law of energy evolution considering viscoelasticity of rock under different stress paths was obtained” and “The concept of ultimate damage energy and its method of calculation were proposed”. As Reviewer suggested that we stated clearly and emphasized this in Abstract and Conclusion (4).
In order to improve the test procedure, acoustic emission monitoring was carried out at the same time to real-time monitor the Microscopic defects of rock materials. In lines 84-85 under the "track changes", we have supplemented this.
Thank you for pointing out this problem in manuscript. It is not reasonable a generalization of that effect. Through testing sandstone and granite, the Conclusion on rock hardening is only applicable to sandstone and granite at present. We corrected this in the Conclusion and replaced all “strengthening” with “hardening” in the Manuscript.
Point 4: Have you any indication of locations of fracture initiation based on acoustic emission sources in the specimens?
Response 4: We gratefully appreciate for your valuable suggestion. We have added AE source locating in Figure 9, Figure 10 and Figure 11. The evolution of AE locating points was analyzed, which not only points out the crack initiation, but also points out the crack propagation and coalescence.
Point 5: Are your estimation and calulculations of energy referred to the whole volume of the specimen? Is it possible for an estimation of energy evolution at the fracture initiation locations?
Response 5: We gratefully appreciate for your valuable suggestion. The estimation and calculation of energy refers to the whole volume of the specimen? It is a very cutting-edge and valuable research that the estimation of energy evolution at the fracture initiation locations. we will conduct research on it in the next step.
Point 6: Improve the conclusions discussing what is your contribution.
Response 6: Thank you for the above suggestions. We have made a comprehensive improvement on the Conclusion (4). We especially emphasized the proposal of the concept and calculation method of ultimate damage energy, which can solve the problem that the damage energy at the peak strength cannot be calculated.
See the attachment for details

Reviewer 3 Report
The manuscript 'Energy Evolution and Damage Characteristics of Rock Materials under Different Cyclic Loading and Unloading Paths' can be considered for publication when the following points are clarified:
-The following sentence ' If the viscoelasticity of rock is not taken into account and the dissipated energy is generally defined as the energy causing rock failure, the calculation result of damage variable will be too high' from Introduction should be elaborated more precisely. Why would you include the viscoelasticity in rock failure in this case? Also viscosity of liquids is mentioned in the previous sentence? Is this the case for liquid saturated rocks?
-It seems that in Figures 13 and 14 input energy is not equal the sum of elastic, damping and damage energy (input != elastic + damping + damage). Can you elaborate why is this not the case? Is the 2nd law of thermodynamics violated here?
-You should elaborate how is your energy split different than traditional theories and also cite traditional methods for computing the energies. You can find some information in 'Crack propagation in dynamics by embedded strong discontinuity approach: Enhanced solid versus discrete lattice model, CMAME, 340, 480-499, 2018'. Damage models and corresponding energy computation for traditional damage models should be cited as well.
Author Response
Point 1: The following sentence ' If the viscoelasticity of rock is not taken into account and the dissipated energy is generally defined as the energy causing rock failure, the calculation result of damage variable will be too high' from Introduction should be elaborated more precisely. Why would you include the viscoelasticity in rock failure in this case? Also viscosity of liquids is mentioned in the previous sentence? Is this the case for liquid saturated rocks?
Response 1: Thank you so much for your careful check. In lines 63-65 under the "track changes", we rewrote this sentence according to the Reviewer’s suggestion. According to References [23-25], the non-dry rock material has the viscoelastic, including particle friction and liquid viscosity. Thus, the viscoelasticity should be considered. We add this reason in lines 62-64 under the "track changes". It is worth noting that liquid saturated rock is also estimated to have a nonlinear hysteresis effect due to the influence of liquid viscosity. This is a valuable suggestion, which we will investigate liquid saturated rocks in the next step. In lines 62-63 under the "track changes", we emphasize that this the case is only applicable to non-dry rocks.
Point 2: It seems that in Figures 13 and 14 input energy is not equal the sum of elastic, damping and damage energy (input != elastic + damping + damage). Can you elaborate why is this not the case? Is the 2nd law of thermodynamics violated here?
Response 2: This is a problem of the picture display, and we are very sorry about it. To prevent symbols and lines from overlapping with the coordinate axis in figure, we set the starting value of y-axis to -0.5. Therefore, the value of damage energy appears to be somewhat large. Energy calculation conforms to the 2nd law of thermodynamics. We attach the data from Figures 13 and 14 for easy check by reviewers.
Figure 13(a)
Figure 13(b)
Figure 14(a)
Figure 14(b)
"Notes: see the attachment for figure"
Point 3: In Section 3.2, the expression of modulus is ambiguous. The concepts of modulus in uniaxial compression and modulus in cyclic loading and unloading should be distinguished so that readers can understand them clearly.
Response 3: We appreciate your good advice. In Section 3.2, we have made a comprehensive modification to distinguish the deformation modulus of uniaxial compression and the elastic modulus of cyclic loading and unloading.
Point 4: You should elaborate how is your energy split different than traditional theories and also cite traditional methods for computing the energies. You can find some information in 'Crack propagation in dynamics by embedded strong discontinuity approach: Enhanced solid versus discrete lattice model, CMAME, 340, 480-499, 2018'. Damage models and corresponding energy computation for traditional damage models should be cited as well.
Response 4: We gratefully appreciate for your valuable suggestion. In Section 4.1 and 4.2, We have elaborated that the traditional theories and our energy split respectively. In lines 276 and 359 under the "track changes", as Reviewer suggested that we added and adjusted relevant references
"See the attachment for details"

Round 2
Reviewer 1 Report
In my opinion, this manuscript has been revised according to the opinions and suggestions of the reviewers. Please make a re-evaluation and consideration and then accept for publication.
Reviewer 2 Report
Accept
Reviewer 3 Report
The authors responded correctly.